# Stationary Trend in Elevated Serum Alpha-Fetoprotein Level in Hepatocellular Carcinoma Patients

**DOI:** 10.3390/cancers15041222

**Published:** 2023-02-15

**Authors:** Yi-Hao Yen, Kwong-Ming Kee, Wei-Feng Li, Yueh-Wei Liu, Chih-Chi Wang, Tsung-Hui Hu, Ming-Chao Tsai, Chih-Yun Lin

**Affiliations:** 1Division of Hepatogastroenterology, Department of Internal Medicine, Kaohsiung Chang Gung Memorial Hospital and Chang Gung University College of Medicine, Kaohsiung 833, Taiwan; 2Liver Transplantation Center, Department of Surgery, Kaohsiung Chang Gung Memorial Hospital, Kaohsiung 833, Taiwan; 3Biostatistics Center, Kaohsiung Chang Gung Memorial Hospital, Kaohsiung 833, Taiwan

**Keywords:** alpha-fetoprotein, hepatocellular carcinoma, hepatitis B virus, hepatitis C virus

## Abstract

**Simple Summary:**

In this study, we demonstrated that overall 51.2% of patients with hepatocellular carcinoma (HCC) had elevated alpha-fetoprotein (AFP) levels. The proportion of patients with elevated AFP levels was stationary in the period from 2011 to 2020. The proportion of patients with Barcelona Clinic Liver Cancer classification (BCLC) stages 0–A HCC decreased from 2011 to 2020, whereas the proportion of patients with non-HBV- and non-HCV (NBNC)-HCC increased in the same period. Furthermore, the proportion of patients with early-stage HCC (i.e., BCLC stages 0–A) was lower for NBNC-HCC than for HBV- or HCV-related HCC. Advanced tumor stage, severe underlying liver disease, viral etiology, and female gender are associated with elevated AFP levels in HCC patients.

**Abstract:**

A recent study from the US showed a decreasing trend in the elevated serum alpha-fetoprotein (AFP) level (i.e., ≥20 ng/mL) in hepatocellular carcinoma (HCC) patients at the time of diagnosis. Furthermore, advanced tumor stage and severe underlying liver disease were associated with elevated AFP levels. We aimed to evaluate this issue in an area endemic for hepatitis B virus (HBV). Between 2011 and 2020, 4031 patients were newly diagnosed with HCC at our institution. After excluding 54 patients with unknown AFP data, the remaining 3977 patients were enrolled in this study. Elevated AFP level was defined as ≥20 ng/mL. Overall, 51.2% of HCC patients had elevated AFP levels; this proportion remained stationary between 2011 and 2020 (51.8% vs. 51.1%). Multivariate analysis showed that female gender (odds ratio (OR) = 1.462; *p* < 0.001), tumor size per 10 mm increase (OR = 1.155; *p* < 0.001), multiple tumors (OR = 1.406; *p* < 0.001), Barcelona Clinic Liver Cancer stages B–D (OR = 1.247; *p* = 0.019), cirrhosis (OR = 1.288; *p* = 0.02), total bilirubin > 1.4 mg/dL (OR = 1.218; *p* = 0.030), and HBV- or hepatitis C virus (HCV)-positive status (OR = 1.720; *p* < 0.001) were associated with elevated AFP levels. In conclusion, a stationary trend in elevated serum AFP level in HCC patients has been noted in the past 10 years. Advanced tumor stage, severe underlying liver disease, viral etiology, and female gender are associated with elevated AFP levels in HCC patients.

## 1. Introduction

Hepatocellular carcinoma (HCC) is one of the leading causes of cancer-related death worldwide [1]. A meta-analysis showed that HCC surveillance is associated with significant improvements in early-stage tumor detection, the receipt of curative therapy, and survival of cirrhotic patients [2]. The American Association for the Study of Liver Diseases (AASLD) guideline recommends HCC surveillance for high-risk populations. The modality recommended for surveillance is ultrasound with or without an alpha-fetoprotein (AFP) serum assay [3]. Ultrasound with an AFP serum assay is recommended for surveillance because a meta-analysis demonstrated that ultrasound alone had low sensitivity in detecting early-stage tumor in cirrhotic patients. The combination of AFP serum assay and ultrasound significantly increases the sensitivity of tumor detection [4]. Currently, the AASLD guideline recommends diagnostic multiphasic magnetic resonance imaging (MRI)/computed tomography (CT) for further evaluation when the AFP level is ≥20 ng/mL on surveillance [3].

Multiple factors, including advanced tumor stage and viral etiology of chronic liver disease, are associated with elevated AFP levels in HCC patients [5]. A recent study from the US found a downtrend in the percentage of HCC cases with elevated AFP levels at the time of diagnosis from 2010 to 2017 in a large cohort from the National Cancer Database. Elevated AFP was defined as ≥20 ng/mL. Furthermore, advanced tumor stage and severe underlying liver disease were associated with elevated AFP levels. The authors suggested that these changes in AFP values at HCC diagnosis were possibly related to the increasing trend in early-stage tumor detection and the shift from viral (i.e., hepatitis B virus (HBV) or hepatitis C virus (HCV)) to nonviral etiology. However, data on the etiology of liver disease were unavailable in the database analyzed in the study [6].

Approximately 90% of HCC cases are associated with a known underlying etiology [7]. In East Asia, the major risk factor is HBV, whereas, in the Western world, it is HCV [7]. The risk of HCC attributed to HCV infection has largely decreased owing to the eradication of the virus with direct-acting antiviral (DAA) agents [8]. Nonalcoholic fatty liver disease (NAFLD), which is usually associated with obesity, metabolic syndrome, or diabetes mellitus, is becoming the fastest growing etiology of HCC, not only in Western countries [9], but also in Asia [10].

Due to the different etiologies of HCC in the East and the West and viral etiology being associated with elevated AFP levels in HCC patients [5], we aimed to evaluate whether there is a downtrend in the percentages of HCC cases with elevated AFP levels at the time of diagnosis and the factors associated with elevated AFP levels in HCC patients in a country from East Asia, where the leading etiology of HCC is HBV.

## 2. Materials and Methods

The study was conducted according to the guidelines of the Declaration of Helsinki and approved by the Institutional Review Board of Kaohsiung Chang Gung Memorial Hospital (reference number: 202201189B0; date of approval: 8 August 2022).

The Institutional Review Board of Kaohsiung Chang Gung Memorial Hospital waived the need for informed consent due to the retrospective and observational nature of the study design. Data were extracted from Kaohsiung Chang Gung Memorial Hospital’s HCC registry database, which holds prospectively collected and annually updated data.

From 2011 to 2020, 4031 patients were newly diagnosed with HCC at the institution. After excluding 54 patients with unknown AFP data, the remaining 3977 patients were enrolled in this study.

### 2.1. Variables of Interest

Patient demographics, tumor size and number, clinical tumor–node–metastasis (TNM) stage (seventh edition of the American Joint Committee on Cancer (AJCC)) [11], Barcelona Clinic Liver Cancer classification (BCLC) stage [12], AFP level, cirrhosis, Child–Pugh class [13], creatinine, bilirubin, international normalized ratio (INR), hepatitis B surface antigen (HBsAg), anti-HCV antibody, alcohol use disorder (AUD), and HCC diagnostic method (i.e., clinical vs. pathological diagnosis) were prospectively collected from the HCC registry data. Infection with HBV was defined as being HBsAg-positive. Infection with HCV was defined as being anti-HCV-antibody-positive, irrespective of viremia. An individual with AUD was defined as a habitual drinker. Demographic information included age, gender, height, and weight. Tumor size was determined according to the results of pathological examination of patients who underwent surgery, whereas it was determined according to the findings of imaging in patients who underwent nonsurgical treatments. Tumor number (solitary vs. multiple) was determined from the findings of imaging. The presence of cirrhosis was indicated by an Ishak score [14] of 5 or 6 in patients who underwent surgery, whereas it was determined according to the findings of imaging in patients who underwent nonsurgical treatments. Cirrhosis was indicated in imaging by small liver size, nodular liver surface, presence of regeneration nodules, left and right lobe liver volume redistribution, etc. [15]. The BCLC stages according to the original version and BCLC stage A were defined within Milan criteria [16].

The raw data for the cohort involved in this study are available via the following digital object identifier: https://www.dropbox.com/scl/fi/6hzj7a3hrqhp5yu4i17qu/afp-trend.xlsx?dl=0&rlkey=srqyaz7t1sqsli6heaoig4bk8 (accessed on 1 January 2023). 

### 2.2. Statistical Analysis

Variables are presented as number and percentage or median and interquartile range. The Chi-square test was used to compare categorical variables. Mann–Whitney *U* test was used to compare continuous variables. Whether there was an increasing or decreasing trend of BCLC stages 0–A or non-HBV- and non-HCV (NBNC)-HCC according to the year of HCC diagnosis was examined for a linear trend using the Chi-square test. Univariate analyses were conducted to explore the association between elevated AFP levels and clinical variables. Variables with *p*-values ≤ 0.1 in univariate analyses were included in a multivariate logistic regression analysis. To avoid collinearity, we examined the correlation between two independent variables using Spearman’s correlation test. If two independent variables had a correlation coefficient above 0.5, then they were determined to be highly correlated with each other and thus, collinear. In this case, we only chose one of the variables for multivariate analysis. In this analysis, we used the following cutoff values for continuous variables: for bilirubin, the upper limit of the normal range (i.e., 1.4 mg/dL); for creatinine, the upper limit of the normal range (i.e., 1.2 mg/dL); and for INR, the upper limit of the normal range (i.e., 1.2). Relative risks are presented as odds ratio (OR) with a 95% confidence interval (CI). To compare with a recent US study [6], we used the same method adopted in that study [6] to interpret the temporal trend of elevated AFP levels. We estimated the percentage of elevated AFP using marginal effects (i.e., the average predicted probability) from a logistic regression model [17]. All statistical analyses were performed using SPSS version 22.0 and R statistical software version 4.0.5. Two-tailed significance values were applied, and the level of statistical significance was defined as *p* < 0.05.

## 3. Results

### 3.1. Trend in Elevated AFP Levels at the Time of HCC Diagnosis

Overall, 2036 (51.2%) patients with HCC had elevated AFP levels. Between 2011 and 2020, the proportion of patients with elevated AFP levels was stationary in the case of all patients (51.8% [95% CI = 49.4–54.1%] in 2011 vs. 51.1% [95% CI = 48.9–53.3%] in 2020); patients in AJCC stage 1 (40.3% [95% CI = 37.5–43.1%] in 2011 vs. 39.7% [95% CI = 37.0–42.3%] in 2020); patients in AJCC stage 2 (45.4% [95% CI = 41.5–49.3%] in 2011 vs. 44.7% [95% CI = 40.9–48.6%] in 2020); patients in AJCC stage 3 (72.6% [95% CI = 68.8–76.3%] in 2011 vs. 71.9% [95% CI = 68.3–75.5%] in 2020); and patients in AJCC stage 4 (79.8% [95% CI = 74.3–85.3%] in 2011 vs. 79.2% [95% CI = 73.7–84.6%] in 2020) (Figure 1).

### 3.2. Trends in Early-Stage Tumor Prevalence and Non-Viral Etiology at the Time of HCC Diagnosis

The proportion of patients with early-stage tumor (i.e., BCLC stages 0–A) decreased in the period from 2011 to 2020 (54.2% vs. 42.5%, *p* < 0.001) (Figure 2). The proportion of patients with non-HBV and non-HCV (NBNC)-HCC increased between the years 2011 and 2020 (from 20.4% to 28.7%, *p* < 0.001) (Figure 3).

### 3.3. Patients’ Characteristics Categorized by AFP Level

Compared to patients with normal AFP levels, a smaller proportion of patients with elevated AFP were male (*p* = 0.02), had a pathological diagnosis of HCC (*p* < 0.001), were in AJCC stage 1 or 2 HCC (*p* < 0.001), had a solitary tumor (*p* < 0.001), were in BCLC stage 0 or A HCC (*p* < 0.001), were in Child–Pugh class A (*p* < 0.001), and had a low body mass index (BMI) (*p* < 0.001). Furthermore, patients with elevated AFP levels had larger tumors (*p* < 0.001), a higher total bilirubin level (*p* < 0.001), and a higher INR (*p* < 0.001) and a higher proportion of them were cirrhotic (*p* = 0.001) and HBsAg-positive (*p* = 0.002). However, there were no significant differences in age, creatinine level, proportion with AUD, and proportion with anti-HCV-antibody-positive status between the two groups (Table 1).

### 3.4. Variables Associated with Elevated AFP Level

Univariate analysis showed that the following variables were associated with elevated AFP levels: tumor size per 10 mm increase (OR = 1.167; 95% CI = 1.146–1.189; *p* < 0.001); using AJCC stage 1 as the reference, AJCC stage 2 (OR = 1.202; 95% CI = 1.013–1.426; *p* = 0.035), AJCC stage 3 (OR =3.909; 95% CI = 3.270–4.674; *p* < 0.001), and AJCC stage 4 (OR = 6.071; 95% CI = 4.553–8.096; *p* < 0.001); multiple tumors (OR = 2.076; 95% CI = 1.818–2.371; *p* < 0.001); using BCLC stages 0–A as the reference, BCLC stages B–D (OR = 2.640; 95% CI = 2.317–3.009; *p* < 0.001); cirrhosis (OR = 1.275; 95% CI = 1.110–1.465; *p* = 0.001); Child–Pugh class B or C (OR = 1.525; 95% CI = 1.287–1.806; *p* < 0.001); total bilirubin > 1.4 mg/dL (OR = 1.454; 95% CI = 1.262–1.676; *p* < 0.001); INR > 1.2 (OR = 1.238; 95% CI = 1.002–1.530; *p* = 0.048); and HBV- or HCV-positive status (OR = 1.355; 95% CI = 1.160–1.583; *p* < 0.001). Because of the strong correlation between AJCC and BCLC stages (correlation coefficient = 0.658, *p* < 0.001), we only selected the BCLC stage for multivariate analysis to avoid collinearity (Table 2).

Multivariate analysis showed that the following variables were associated with elevated AFP levels: female gender (OR = 1.462; 95% CI = 1.256–1.701; *p* < 0.001); tumor size per 10 mm increase (OR = 1.155; 95% CI = 1.127–1.183; *p* < 0.001); multiple tumors (OR = 1.406; 95% CI = 1.205–1.641; *p* < 0.001); BCLC stages B–D (OR = 1.247; 95% CI = 1.037–1.500; *p* = 0.019); cirrhosis (OR = 1.288; 95% CI = 1.099–1.509; *p* = 0.02); total bilirubin > 1.4 mg/dL (OR = 1.218; 95% CI = 1.020–1.455; *p* = 0.030); and HBV- or HCV-positive status (OR = 1.720; 95% CI = 1.451–2.038; *p* < 0.001) (Table 2).

### 3.5. Proportion of BCLC Stages 0–A Patients in NBNC-HCC vs. HBV- or HCV-Related HCC

Of the 870 patients with NBNC-HCC, 315 (36.2%) were in BCLC stages 0–A, 529 (60.8%) were in BCLC stages B–D, and 26 (0.3%) were of unknown BCLC stage. Of the 3107 HBV- or HCV-positive patients, 1607 (51.7%) were in BCLC stages 0–A, 1439 (46.3%) were in BCLC stages B–D, and 61 (0.2%) were of unknown BCLC stage. A significantly lower proportion of NBNC-HCC patients were in BCLC stages 0–A compared to HBV- or HCV-related HCC patients (*p* < 0.001).

## 4. Discussion

In this study, we demonstrated that overall 51.2% of patients with HCC had elevated AFP levels. The proportion of patients with an elevated AFP level was stationary in the period from 2011 to 2020. The proportion of patients with BCLC stages 0–A HCC decreased from 2011 to 2020, whereas the proportion of patients with NBNC-HCC increased in the same period. Furthermore, the proportion of patients with early-stage HCC (i.e., BCLC stages 0–A) was lower for NBNC-HCC than for HBV- or HCV-related HCC. Our previous study reported that the HCC cases in our institution accounted for 9.8% of the total cases at the national level [18].

Independent factors associated with elevated AFP levels included female gender, increased tumor size, multiple tumors, BCLC stages B–D, cirrhosis, total bilirubin > 1.4 mg/dL, and viral etiology. Advanced tumor stage and viral etiology were associated with elevated AFP levels. Between 2011 and 2020, the proportion of patients with NBNC-HCC increased (which would have led to a decreasing trend in AFP elevation), which was counterbalanced by the decreased proportion of patients with BCLC stages 0–A HCC (which would have led to an increasing trend in AFP elevation). Ultimately, the proportion of patients with elevated AFP levels was stationary between the years 2011 and 2020 in this study. Furthermore, the decreasing proportion of patients with BCLC stages 0–A HCC during this period may be due to the concurrent increase in the proportion of patients with NBNC-HCC, because the BCLC stages 0–A were less frequently found in NBNC-HCC patients compared to HBV- or HCV-related HCC patients, a result that agrees with the findings of a previous study [19]. In that study, patients with NBNC-HCC presented with larger tumors and at later stages of disease compared to patients with virus-related HCC [19]. This result may be due to the low rate of HCC surveillance in NBNC-HCC patients; furthermore, a significant proportion of NBNC-HCC patients could be NAFLD-related cases [20].

A recent study from the US reported an overall 62.6% of HCC patients with elevated AFP levels (i.e., ≥20 ng/mL) at the time of diagnosis. Between 2010 and 2017, there was a decline in the percentage of HCC patients with elevated AFP levels (68.2% vs. 57.5%). Furthermore, the decline was most evident among patients with early-stage tumors (i.e., seventh edition AJCC stage 1), from 55.7% in 2010 to 40.7% in 2017. However, the authors did not investigate the potential cause of these results. They assumed that these results were likely due to the increasing trend in early-stage tumor detection and the shift from viral to nonviral etiology [6]. In contrast, in this study, an overall 51.2% of patients with HCC had elevated AFP levels. The proportion of patients with elevated AFP levels was stationary in the period from 2011 to 2020 for all patients and across different AJCC stages in the present study. However, it is unclear why there is a discrepancy between this study and the US study [6].

Previous studies have shown that female gender, viral etiology, severe underlying liver disease, and advanced tumor stage are independently associated with elevated levels of AFP [5,6], findings that are compatible with those of the present study. Interestingly, female gender is associated with elevated AFP levels in the present study and previous large-scale studies [5,6]. However, the underlying mechanism is still unknown. It is well known that severe liver disease and advanced tumor stage are associated with elevated AFP levels [5,6,21]. The link between viral etiology and elevated AFP may be due to the former’s association with cirrhosis. A previous study demonstrated that NAFLD is the leading cause of non-cirrhotic HCC [22].

In Taiwan, HBV is the leading etiology of HCC [18]. In 1984, Taiwan established a universal HBV vaccination program for newborns [23,24]. HBV vaccination has reduced the incidence of HCC in children and adolescents [23,24]. Antiviral therapy for HBV and HCV has been widely used in Taiwan since 2003. Antiviral therapies reduce the incidence of HBV- and HCV-related HCC [25,26]. Consequently, the incidence of HBV- and HCV-related HCC was expected to decrease. Indeed, of the 3843 HCC patients from five medical centers in Taiwan enrolled by Chang et al. in their study during 2005–2011, only 10.7% had NBNC-HCC [27], in contrast to 20.4% in 2011 and the gradual rise to 28.7% in 2020 in the present study. Therefore, we infer that the universal HBV vaccination program for newborns and the widely employed antiviral therapy for HBV and HCV have resulted in the decreasing incidence of HBV- and HCV-related HCC in Taiwan.

The etiologies of NBNC-HCC may be AUD and NAFLD; other etiologies, such as autoimmune hepatitis, primary biliary cirrhosis, and primary sclerosing cholangitis, related to chronic liver disease are rare [22]. Patients with NBNC-HCC have an increased risk of metabolic comorbidities [28], which implies that a significant proportion of NBNC-HCC cases could be NAFLD-related.

The incidence of HCC attributed to a nonviral etiology (mainly NAFLD) is rising [29,30]; in addition, it is associated with a decline in the percentage of patients with elevated AFP levels. Other tumor markers more specific to NBNC-HCC are needed for HCC surveillance in this population. Another available tumor marker for HCC surveillance is des-γ-carboxyprothrombin (DCP), which is recommended in clinical practice guidelines by the Japan Society of Hepatology [31]. Previous studies have demonstrated that elevated DCP might be a diagnostic marker for NBNC-HCC [32,33].

Due to the obesity pandemic, the challenge is how to screen for HCC in patients with NAFLD. The American Gastroenterological Association (AGA) guidelines recommend that screening for HCC should be considered for cirrhotic patients due to NAFLD. When the quality of ultrasound is suboptimal for HCC screening (e.g., due to obesity), future surveillance should be performed using CT or MRI, with or without determining the AFP level, every 6 months [34]. However, the cost-effectiveness of HCC surveillance, if CT or MRI replaces ultrasound, remains unknown [34].

Modern molecular biology-based technologies (e.g., liquid biopsy) hold considerable promise for early diagnosis of HCC. To date, there are still no US Food and Drug Administration (FDA)-approved liquid biopsy assays for HCC, mainly due to the lack of survival benefit of such assays [35].

The strength of the present study is the use of a large cohort of patients with HCC with prospectively collected data and limited missing data. However, the study has several limitations. First, we did not use the Alcohol Use Disorders Inventory Test (AUDIT) [36] or AUDIT-C [37,38] (which is recommended by the European Association for the Study of the Liver (EASL) guidelines [39]) to screen HCC patients for AUD. In contrast, we reviewed medical records and used habitual drinking to define AUD, which may underestimate the prevalence of AUD in the present study. Second, the study lacked data on antiviral therapy for HBV and HCV, which can affect AFP levels. Previous studies have shown that the cutoff values of AFP for HCC surveillance are lower in HBV-related cirrhosis patients receiving nucleos(t)ide analogue therapy [40] and HCV-related cirrhosis patients treated with DAAs [41]. Third, the study lacked data on HCV ribonucleic acid (RNA). A proportion of patients with anti-HCV-antibody-positive status could have experienced a past episode of resolved HCV infection. Fourth, etiologies other than HBV, HCV, and AUD were unavailable. Furthermore, we also did not have data on hepatic steatosis and metabolic comorbidities [42]. Therefore, we could not define NAFLD in the present study. Finally, this is a retrospective study.

## 5. Conclusions

In the past 10 years, a stationary trend in elevated serum AFP levels in HCC patients was noted in this cohort from an HBV-endemic area. This result may be due to the proportion of patients with early-stage HCC decreasing and the proportion of patients with NBNC-HCC increasing during this period. Furthermore, advanced tumor stage, severe underlying liver disease, female gender, and viral etiology were associated with elevated AFP in HCC patients.

## Figures and Tables

**Figure 1 cancers-15-01222-f001:**
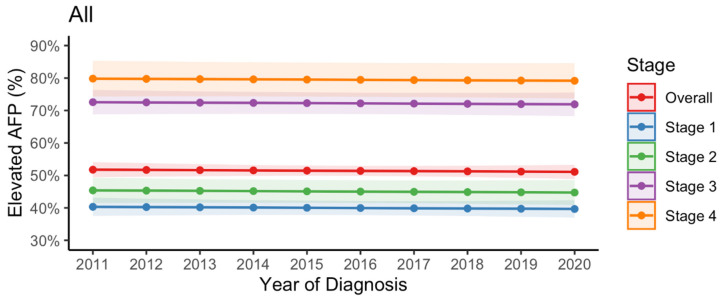
Trend in elevated serum alpha-fetoprotein levels (i.e., ≥20 ng/mL) at the time of hepatocellular carcinoma diagnosis in all patients and at different stages (seventh edition American Joint Committee on Cancer).

**Figure 2 cancers-15-01222-f002:**
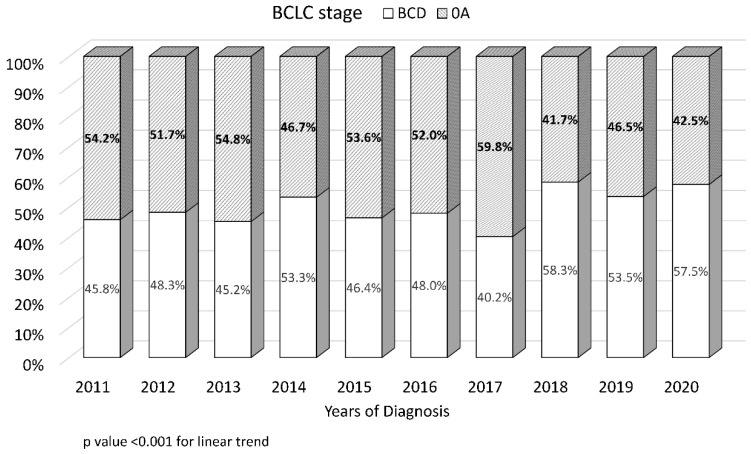
Trend in early-stage tumor at the time of hepatocellular carcinoma diagnosis. Early-stage tumor was defined as Barcelona Clinic Liver Cancer classification stages 0–A.

**Figure 3 cancers-15-01222-f003:**
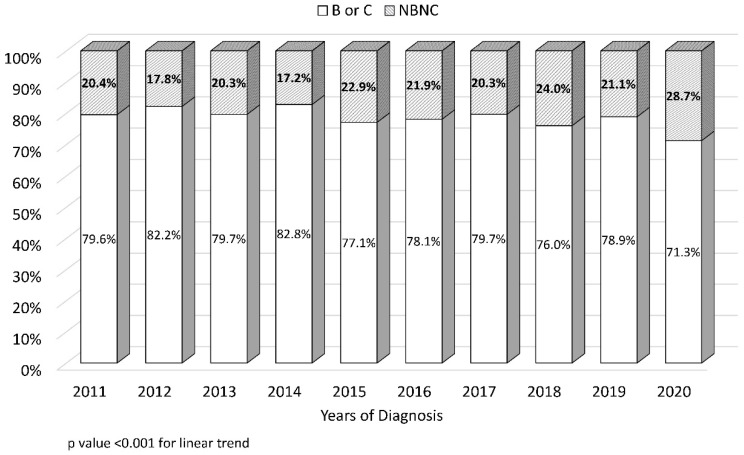
Trend in nonviral etiology (i.e., patients negative for both hepatitis B and C viruses) at the time of hepatocellular carcinoma diagnosis.

**Table 1 cancers-15-01222-t001:** Patients’ characteristics categorized by alpha-fetoprotein level.

Characteristic	Elevated AFP (≥20 ng/mL), *n* = 2036	Normal AFP (<20 ng/mL), *n* = 1941	*p*
Age (years)	63 (55–71)	63 (56–71)	0.286
Male	1437 (70.6%)	1434 (73.9%)	0.02
BMI (kg/m^2^)	24.2 (22.1–26.9)	24.9 (22.5–27.8)	<0.001
Diagnosis method			<0.001
Clinical diagnosis	896 (44.0%)	593 (30.6%)	
Pathology diagnosis	1140 (56.0%)	1348 (69.4%)	
AUD			0.460
Yes	260 (12.8%)	225 (11.6%)	
No	1762 (86.5%)	1705 (87.8%)	
Not available	11 (0.6%)	14 (0.7%)	
HBsAg			0.002
Positive	1006 (49.4%)	866 (44.6%)	
Negative	1030 (50.6%)	1075 (55.4%)	
Anti-HCV			0.327
Positive	741(36.4%)	674 (34.7%)	
Negative	1295 (63.6%)	1266 (65.2%)	
Not available	0	1 (0.1%)	
Cirrhosis			0.001
Yes	1469 (72.2%)	1294 (66.7%)	
No	561 (27.6%)	640 (33.0%)	
Not available	7 (0.4%)	6 (0.3%)	
Child–Pugh class			<0.001
A	1587 (77.9%)	1627 (83.80%)	
B	345 (16.9%)	235 (12.1%)	
C	70 (3.4%)	51 (2.6%)	
Not available	34 (1.7%)	28 (1.4%)	
Creatinine (mg/dL)	1.0 (0.8–1.3)	1.0 (0.8–1.3)	0.731
Total bilirubin (mg/dL)	1.1 (0.8–1.7)	1.0 (0.7–1.4)	<0.001
INR	1.0 (1.0–1.1)	1.0 (1.0–1.1)	<0.001
Tumor size (mm)	46 (26–98)	28 (20–45)	<0.001
Tumor number			<0.001
Single	1064 (52.3%)	1360 (70.1%)	
Multiple	972 (47.7%)	581 (29.9%)	
7th edition AJCC stage			<0.001
1	788 (38.7%)	1168 (60.2%)	
2	340 (16.7%)	414 (21.3%)	
3	617 (30.3%)	233 (12.0%)	
4	279 (13.7%)	72 (3.7%)	
Not available	12 (0.6%)	54 (2.8%)	
BCLC stage			<0.001
0	206 (10.1%)	293 (15.1%)	
A	566 (27.3%)	867 (44.7%)	
B	446 (21.9%)	362 (18.7%)	
C	676 (33.2%)	279 (14.4%)	
D	124 (6.1%)	81(4.2%)	
Not available	28 (1.4%)	59 (3.0%)	

AFP, alpha-fetoprotein; BMI, body mass index; AUD, alcohol use disorder; HBsAg, hepatitis B surface antigen; anti-HCV, anti-hepatitis C virus antibody; INR, international normalized ratio; AJCC, American Joint Committee on Cancer; BCLC, Barcelona Clinic Liver Cancer.

**Table 2 cancers-15-01222-t002:** Univariate and multivariate analyses of factors associated with elevated AFP levels.

Variable	Univariate	*p*	Multivariate	*p*
	OR (95% CI)		OR (95% CI)	
Age (per 10 years)	0.955 (0.904–1.010)	0.105		
Female vs. male	1.20 (1.04–1.39)	0.011	1.462 (1.256–1.701)	<0.001
AUD	1.077 (0.887–1.306)	0.454		
HBsAg or anti-HCV-positive	1.355 (1.160–1.583)	<0.001	1.720 (1.451–2.038)	<0.001
Cirrhosis	1.275 (1.110–1.465)	0.001	1.288 (1.099–1.509)	0.02
Child–Pugh class B or C vs. A	1.525(1.287–1.806)	<0.001	0.964 (0.770–1.205)	0.745
Creatinine >1.2 mg/dL	0.983 (0.849–1.138)	0.821		
Total bilirubin >1.4 mg/dL	1.454 (1.262–1.676)	<0.001	1.218 (1.020–1.455)	0.030
INR >1.2	1.238 (1.002–1.530)	0.048	0.897 (0.687–1.172)	0.425
Tumor size, per 10 mm increase	1.167 (1.146–1.189)	<0.001	1.155 (1.127–1.183)	<0.001
Multiple tumors	2.076 (1.818–2.371)	<0.001	1.406 (1.205–1.641)	<0.001
BCLC stage (O–A as reference)				
B–D	2.640 (2.317–3.009)	<0.001	1.247 (1.037–1.500)	0.019
7th edition AJCC Stage 1 as reference				
Stage 2	1.202 (1.013–1.426)	0.035		
Stage 3	3.909 (3.270–4.674)	<0.001		
Stage 4	6.071 (4.553–8.096)	<0.001		

AFP, alpha-fetoprotein; AUD, alcohol use disorder; HBsAg, hepatitis B surface antigen; anti-HCV, anti-hepatitis C virus antibody; INR, international normalized ratio; AJCC, American Joint Committee on Cancer; BCLC, Barcelona Clinic Liver Cancer.

## Data Availability

The data can be shared up on request.

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
