# Peer review of "Stationary Trend in Elevated Serum Alpha-Fetoprotein Level in Hepatocellular Carcinoma Patients"

_cancers, 2023, doi:10.3390/cancers15041222_

Round 1
Reviewer 1 Report
thank you for the opportunity to review this article.
Can i check:
1) is there data on the number of HCC diagnosed at the national level? this may give readers an idea of the proportion of patients with HCC /whole country vs in your hospital system. This is important to understand whether there maybe other factors that influence the study results.
2) How does the results impact the way that readers /clinicians perform their clinical practice? While the results maybe be interesting as a demographic dataset correlating HCC presentation with AFP - i fail to see how this will affect the clinical management or change practice for clinicians in general.
Author Response
Reviewer 1.
Can i check:
- is there data on the number of HCC diagnosed at the national level? this may give readers an idea of the proportion of patients with HCC /whole country vs in your hospital system. This is important to understand whether there maybe other factors that influence the study results.
Response: Thank you so much for your comments. Our previous study reported that
data regarding patients diagnosed as having HCC from January 2011 to December 2017, were extracted from the Taiwan Cancer Registry (i.e., the national level) and the database of our institution. A total of 43,437 patients with HCC were enrolled from the Taiwan Cancer Registry database, and 4250 patients diagnosed as having HCC in our institution. The HCC cases in our institution accounted for 9.8% of the total cases at the national level [18]. Please see line 212-214
- How does the results impact the way that readers /clinicians perform their clinical practice? While the results maybe be interesting as a demographic dataset correlating HCC presentation with AFP - i fail to see how this will affect the clinical management or change practice for clinicians in general.
Response:
The incidence of HCC attributed to a nonviral etiology (mainly NAFLD) is rising [29,30]; it is also associated with a decline in the percentage of patients with elevated AFP. Other tumor markers more specific to NBNC-HCC are needed for HCC surveillance in this population. Another available tumor marker for HCC surveillance is des-γ-carboxyprothrombin (DCP), which is recommended in clinical practice guidelines by the Japan Society of Hepatology [31]. Previous studies have demonstrated that elevated DCP might be a diagnostic marker for NBNC-HCC [32,33]. Please see line 269-275.

Reviewer 2 Report
In this study the authors attempt to clarify whether the downward trend in AFP levels found in a recent US study may be related to the detection of HCC at early stages as well as the influence of non-viral factors.
The authors show stationary levels of AFP in patients diagnosed between 2011 and 2020, with the highest values found in advanced stages. To clarify whether these values are associated with early stages and viral infections, they show the early stage based on BCLC and the relationship with HBV and HCV, determining an upward trend in HCC not related to viruses and a reduced proportion of patients in early stage in that period. However, the association with viruses continues to be the majority.
This study shows a clear difference from a previous study in the US, which demonstrates a clear difference in trend between eastern and western countries which may be due to genetic and environmental factors such as diet.
The stationary values during the same period in Taiwan may be due to the reduction in early-stage HCC which may be offset by the reduction in viral influence.
This study shows the stage and presence of viral infection, which justifies, at least in part, the stationary levels of AFP.
Although the study should be completed in other contexts of liver injury as well as genetic and environmental factors, this study should be published since they partially justify the data obtained.
Author Response
Thank you so much for your comments.

Reviewer 3 Report
Yen and al proposed an interesting study above AFP evolution in the last 10 years. They show a stable trend.
The paper is wel-documented. Authors report limitations, notbly in causes.
Nevertheless, some results would be interesting to know: it seems that risk factors of elevated AFP concern all patients, but it would be more interesting to konow if there is a trend in these risk factors. Number of HCC diagnosed per year would be interesting too.
Finally, comparison between other results, particularly in elevated AFP prevalence is important: please mention for example prevalence in USA.
Author Response
Reviewer 3
Yen and al proposed an interesting study above AFP evolution in the last 10 years. They show a stable trend.
The paper is wel-documented. Authors report limitations, notbly in causes.
Nevertheless, some results would be interesting to know: it seems that risk factors of elevated AFP concern all patients, but it would be more interesting to konow if there is a trend in these risk factors. Number of HCC diagnosed per year would be interesting too.
Response: Thank you so much for your comments. Due to there were nine risk factors associated with elevated AFP level (please see table 2, univariate analysis). It would be too crowded if we put all these figures in the text.
There were 373 patients diagnosed in 2011, 354 patients diagnosed in 2012, 359 patients diagnosed in 2013, 326 patients diagnosed in 2014, 367 patients diagnosed in 2015,338 patients diagnosed in 2016, 325 patients diagnosed in 2017,530 patients diagnosed in 2018,497 patients diagnosed in 2019 and 798 patients diagnosed in 2020.
Finally, comparison between other results, particularly in elevated AFP prevalence is important: please mention for example prevalence in USA.
Response: A recent study from the US reported an overall 62.6% of HCC patients with elevated AFP levels (i.e., ≥ 20ng/ml) at the time of diagnosis. In contrast, in this study, an overall 51.2% of patients with HCC had elevated AFP levels. Please see line 232-240.

Round 2
Reviewer 3 Report
The authors gave their appropriate answer and corrected the manuscript.